# Overview on Aneuploidy in Childhood B-Cell Acute Lymphoblastic Leukemia

**DOI:** 10.3390/ijms24108764

**Published:** 2023-05-15

**Authors:** Kinga Panuciak, Emilia Nowicka, Angelika Mastalerczyk, Joanna Zawitkowska, Maciej Niedźwiecki, Monika Lejman

**Affiliations:** 1Student Scientific Society, Independent Laboratory of Genetic Diagnostics, Medical University of Lublin, 20-093 Lublin, Poland; kinga.panuciak26@gmail.com (K.P.); e.nowicka22@gmail.com (E.N.); ngmastaler@gmail.com (A.M.); 2Department of Pediatric Hematology, Oncology and Transplantology, Medical University of Lublin, Gębali 6, 20-093 Lublin, Poland; joanna.zawitkowska@umlub.pl; 3Department of Pediatrics, Hematology and Oncology, Medical University of Gdansk, Debinki 7, 80-211 Gdansk, Poland; maciej.niedzwiecki@gumed.edu.pl; 4Independent Laboratory of Genetic Diagnostics, Medical University of Lublin, 20-093 Lublin, Poland

**Keywords:** B-cell acute lymphoblastic leukemia, aneuploidy, prognosis, chromosomal abnormalities, high hyperdiploidy, hypodiploidy, masked hypodiploid karyotype

## Abstract

Recent years have brought significant progress in the treatment of B-cell acute lymphoblastic leukemia (ALL). This was influenced by both the improved schemes of conventionally used therapy, as well as the development of new forms of treatment. As a consequence, 5-year survival rates have increased and now exceed 90% in pediatric patients. For this reason, it would seem that everything has already been explored in the context of ALL. However, delving into its pathogenesis at the molecular level shows that there are many variations that still need to be analyzed in more detail. One of them is aneuploidy, which is among the most common genetic changes in B-cell ALL. It includes both hyperdiploidy and hypodiploidy. Knowledge of the genetic background is important already at the time of diagnosis, because the first of these forms of aneuploidy is characterized by a good prognosis, in contrast to the second, which is in favor of an unfavorable course. In our work, we will focus on summarizing the current state of knowledge on aneuploidy, along with an indication of all the consequences that may be correlated with it in the context of the treatment of patients with B-cell ALL.

## 1. Introduction

The World Health Organization (WHO) classifies tumors based on scientific data and includes cancers that affect different organ systems. It is used as a benchmark for public health monitoring, cancer registries, research, and diagnosis all around the world. The current series (5th edition) has been created inside a unified relational database system that covers all human cancers for the first time since the classification’s start more than 60 years ago [1,2].

The 5th edition of the World Health Organization Classification of Hematolymphoid Tumors (WHO-HAEM5) classifies some genetic anomalies as subtypes, including high hyperdiploidy, which was not included in previous classifications. Under the 2022 WHO-HAEM 5 classification, precursor B-cell neoplasms are classified based on ploidy changes, such as hyperdiploidy and hypodiploidy, as well as chromosomal rearrangements or the presence of other genetic drivers [3,4,5,6,7] (Figure 1).

In turn, the updated International Consensus Classification (ICC) of B-acute lymphoblastic leukemia (B-ALL) and T-acute lymphoblastic leukemia (T-ALL) includes subtypes that initially appeared in the 2016 WHO classification, and several new entities are being defined [8]. The classification of B-ALL has been altered to additionally subclassify BCR::ABL1-positive B-ALL and most significantly, hypodiploid B-ALL. Next to B-ALL with different gene mutations, the ICC of ALL establishes separate subtypes, such as B-ALL hyperdiploid and B-ALL with hypodiploidy, which is now officially recognized in the ICC and divided into two groups: “B-ALL low hypodiploid” (32–39 chromosomes) and “B-ALL near-haploid” (24–31 chromosomes) [9,10].

Among all ALL cases, B-cell ALL is more characteristic for children (85% of cases), and represents 75% of cases among adults [11]. Within both of these groups of patients, disorders in the form of aneuploidy can be detected—aneuploidy is a term referred to as cells with a total number of chromosomes that is not a multiple of the normal haploid complement and represents a state with an imbalanced karyotype [12,13]. The most characteristic subtypes with aneuploidy are B-cell ALL with hypodiploidy and hyperdiploidy, while aneuploidy in T-cell ALL is much less described. T-ALL occurs more frequently in adults than in children and the WHO-HAEM-5 defines only a subtype of early T-cell precursor lymphoblastic leukemia [8,9,13,14,15,16,17,18]. The only aneuploidy that happens often among T-ALL cases is near-tetraploidy—in the study described by Ceppi et al., 1.4% of patients were diagnosed with near-tetraploid T-ALL [11,19]. Although there is no universally accepted definition of aneuploidy in the literature, scientists strongly urge the field to implement the definition of aneuploidy as copy number alterations (CNAs) that impact either entire chromosome arms or whole chromosomes in order to investigate the significance of aneuploidy in cancer progression and for practical reasons. A terminology that is comparable would improve uniformity among cancer studies [20,21].

As we mentioned, leukemia subtypes are classified now according to the WHO-HAEM-5 and the ICC, but none of them refer to the division or classification of chromosomes—only to leukemia subtypes characterized by genetic changes; therefore, for the purposes of this review article, we made the decision to rely on the 2020 International System of Cytogenetic Nomenclature classification, which is an international up-to-date guide for human cytogenetics, molecular biologists, technicians and students to interpret and communicate human cytogenomic nomenclature, as well as for classifying described aneuploidies. We have decided to divide the ALL subtypes depending on the ploidy of the chromosomes affected: cases with 23–29 chromosomes (hyperhaploidy) count as near-haploid, 30–39 chromosomes (hypodiploidy) as low hypodiploidy and 40–44 chromosomes is classified as high hypodiploidy [17,22]. Cases with 47–57 chromosomes count as hyperdiploidy, and 58-80 ploidy is classified as near-triploidy. For this reason, in our comparison, low hyperdiploidy concerns 47–50 chromosomes, while high hyperdiploidy contains 51–57 chromosomes. Near-triploidy is defined as 58–80 chromosomes and near-tetraploidy is defined as 81–103 chromosomes Table 1.

We systematically reviewed published studies about childhood B-cell ALL. The PubMed and Wiley Online Library databases were searched for studies in humans published or translated in English between 1 January 2000 and 23 March 2023, but focused on the publications from the last 5 years. Additional searches were conducted for congress proceedings [American Society of Clinical Oncology (ASCO), American Society of Hematology (ASH), European Hematology Association (EHA), and European Society for Medical Oncology (ESMO)]. The full texts were obtained for all studies deemed eligible, and for studies whose eligibility was unclear during the title/abstract screening. The full texts were independently screened to confirm which studies should be included.

## 2. Hyperdiploidy

Hyperdiploidy is a term used to describe a specific genetic change that occurs in B-cell acute lymphoblastic leukemia (ALL). In these cases, the number of chromosomes in the affected cells is greater than the normal number of 46 chromosomes, and there are an extra one or more copies of certain chromosomes [23]. Its classic form of hyperdiploidy consists of heterozygous di-, tri- and tetrasomies, while there are only di- and tetrasomies in the non-classical [23]. It was first described over 50 years ago, and it has been divided into two main subgroups with further research [24].

### 2.1. Low Hyperdiploidy

Low hyperdiploidy accounts for approximately 10–15% of diagnosed cases and its incidence increases with age [25,26,27,28]. Chromosomal gains can occur within almost any chromosome, but more often they are observed in the case of chromosomes 8, 10, 21 and X [27,29]. Structural abnormalities occur in approximately 75% of cases and *ETV6*::*RUNX1* translocation is common, especially in the presence of trisomy 21 [30]. The most common variant in this group seems to be patients with 47 chromosomes. This is in line with the findings of Jarosov et al., where five out of seven patients examined had exactly this number of chromosomes. In the context of this study, it is worth mentioning one of the examined patients who died as a result of a relapse of the disease 2 years after its diagnosis. During its analysis, the daughter chromosome 20 associated with the t(9;20) translocation and trisomy 5 of the chromosome were visualized. This trisomy as a single cytogenetic abnormality is rare in ALL and may be associated with a worse prognosis. Therefore, it was concluded that both of the patient’s cytogenetic changes in combination with the present hyperleukocytosis had a cumulative effect on each other in the context of an unfavorable prognosis [31].

Low hyperdiploidy is also associated with shorter survival time and thus unfavorable prognosis, even after intensive treatment [25,28,30,32,33] Table 2. 

This was confirmed in the study by Groeneveld-Krentz et al., where 10-year event-free survival in the case of recurrence in patients with low hyperdiploidy was only 40%. In contrast, in the group of high hyperdiploidy, it was 70% [29]. 

When discussing this aberration, it is worth mentioning the frequency of its occurrence in the context of geographical distribution, which may vary depending on the population. This was obtained by Forestier at al., whose study was conducted on a group of pediatric patients from Nordic countries. The subgroup with the number of chromosomes 47–51 was accounted for 17% out of 787 abnormal ALL cases. A slightly higher percentage of detected cases with this aberration is noticeable compared to data from other studies. However, the basis of this difference has not been finally confirmed. It may result from differences in individual laboratories, exposure to various environmental conditions, as well as geographical variability [34]. This draws attention to the fact that currently there is still insufficient research on the geographical distribution of individual cytogenetic variants of ALL. Although a higher incidence of this disease (up to 90%) is recorded in African and Asian countries, the vast majority of the studies come from American and European centers [35]. When Asian studies do appear, they often involve a smaller research group. One of these is the Indian study by Mazloumi et al., in which low hyperdiploidy was observed in 14.2%. Unfortunately, in their group, this result corresponded to only two patients [36]. For this reason, the study of Lee et al., which draws attention to this issue, seems significant. Although in their results the hyperdiploidy group was not categorized by the exact number of chromosomes, it was noticeable that this aberration was more common in the group of East Asian ancestry compared to the groups of South Asian, African and American ancestry. European origin was omitted as a reference point in this comparison [35]. Thus, it could confirm the data collected by Amare et al., in which low hyperdiploidy was diagnosed in 339 (44%) children of Indian descent [37]. However, the presented problem regarding knowledge about ethnic frequency is not the only one that remains to be investigated. Within low hyperdiploidy, there are still many question marks. Due to the fact that within hyperdiploidy its variant with higher ploidy is more common than the low group, relatively few studies refer only to this group as a separate entity. Undoubtedly, issues such as environmental factors or genetic changes that may co-occur with low hyperdiploidy, as well as indicate it, require additional research. In the context of patients, the outcome after standard treatment compared to other groups would also be important. It is possible that such research could lead to better personalized therapy, thus improving the prognosis.
ijms-24-08764-t002_Table 2Table 2Genetic subtypes according to the ploidy of the chromosome of B-cell acute lymphoblastic leukemia in children. WBC: white blood cells; EFS: event-free survival.Subtype of the ALLFrequency (%)Age (Median)WBC (×10^9^/L)5-Year EFSChromosomes (Loss or Gain)Genetic AlterationsImmunophenotypeReferencesNear-haploidy0.56.2<5025–40%3, 7, 9, 15, 16 and 17*NRAS, FLT3, KRAS, PTPN11, NF1*, histone modifiers, *CREBBP, CDKN2A/B*, histone gene cluster 6p22, *IKZF3, PAG1*CD19, CD22, CD34, cCD79a and TdT[15,28,38,39,40] Low hypodiploidy0.512.9≤2030–50%3, 7, 13, 16, and 17*TP53, CREBBP, RB1, IKZF2, CDKN2A/2B*pro-B ALL phenotype[15,17,28,38,39,40]High hypodiploidy0.54≤2075%7, 9, and 12*CDKN2A, TP53*pro-B ALL phenotype, Common ALL[38,39,41]Low hyperdiploidy10–154≤2050%X, 8, 10, and 21*FLT3, NRAS, KRAS* and *PTPN11*Common ALL[29,32]High hyperdiploidy25–304<1090%X, 4, 6, 8, 10, 14, 17, 18 and 21*CEBPE, ARID5B, PIP4K2A, BMI, GAB2, CREBBP, WHSC1, SUV420H1, SETD2, EZH2, FLT3, NRAS, KRAS, PTPN11*CD9, CD20, CD22, CD58, CD66c, CD86 and CD123[42,43,44,45,46]Near-triploidy0.3–2.14.5≤1594%X, 4, 5, 6, 8, 10, 11, 12, 13, 14, 15, 16, 17, and 21*TP53, ETV6::RUNX1*CD19, CD22[47,48]Near-tetraploidy1.48.6≤1589%X, 3, 5, 7, 8, 10, 11, 12, 17, 18, 19, 20, and 21*ETV6*::*RUNX1*CD19, CD22[48,49,50,51]

### 2.2. High Hyperdiploidy

High hyperdiploidy ALL is defined genetically by extensive aneuploidy, which appears as a non-accidental increase in the number of chromosomes leading to individual trisomies and tetrasomies [52]. It is detected in approximately 25–30% of children with B cell precursor acute lymphoblastic leukemia (BCP-ALL) and constitutes the largest cytogenetic subgroup of this disease [26,53,54]. Frequently it affects young children with B-ALL. It rarely occurs in infants and in T-cell ALL or Burkitt leukemia/lymphoma. The most commonly reported median age is 4 years, which is in line with the results of a study by Sun et al., where most often the disease was diagnosed between 1 and 9 years of age. However, according to other studies, it is rare after the age of 7 [52,55,56].

In the clinical context, low white blood cell (WBC) counts with a median of <10 × 10^9^/L are frequently found in patients at the time of diagnosis. It is indicated that the number of blasts in the blood may be 30–60%; however, data on this subject are deficient. In addition, the occurrence of moderate anemia is described, where the average hemoglobin level varies between 60 and 75 g/L. Thrombocytopenia with a median concentration of <50 × 10^9^/L can be observed in a significant group of patients; however, normal platelet counts have also been reported. Extramedullary leukemia (including mediastinal tumors or central nervous system involvement) is rare in these patients and is not a significant factor. In contrast, a high percentage of blasts (with a median of 95–100%) in the bone marrow is often detected [55]. In flow cytometric analysis, high hyperdiploid cells show significantly higher expression of CD9, CD20, CD22, CD58, CD66c, CD86 and CD123 antigens compared to other ploidy groups. In contrast, CD45 is down-regulated [21]. This fact is used in the case of abnormal expression of CD123, the alpha chain of the interleukin-3 receptor, which is commonly used as an indicative surrogate predictor of flow cytometry for classic leukemias with high hyperdiploidy [23]. In order to compare the ratio of incidence to gender, the study by Sharathkumar et al. should be mentioned. In it, they obtained a male to female ratio of 1.4. This was consistent with some previous studies where high hyperdiploidy predominated in boys. However, this dominance does not seem to be significant, especially that in larger series it is close to 1.0 [57,58,59].

Similary, as in the case of low hyperdiploidy the percentage of its occurrence, may vary depending on the study group. It has been proven that compared to the Western world, the frequency of its existence in people of Asian, African and Native American descent is much lower and amounts to approx. 15–25% of children’s ALL [23]. This is confirmed by studies conducted on the Chinese population, in which the total incidence of high hyperdiploidy in pediatric patients was 17.9%. In terms of gender, it was 16.9% among all surveyed men and 19.6% within the entire female group [60]. Similar results were obtained by Liang et al., who showed high hyperdiploidy (>50 chromosomes) in 14.1% of 433 Taiwanese children with ALL [61]. In turn, Fadoo et al. obtained 10.7%, which corresponds to 34 Pakistani children [62].

Moving on to the characterization at the genetic level, the gains are most common with chromosomes X, 4, 6, 8, 10, 14, 17, 18 and 21, occurring in approximately 75% of cases [33,42,54,55,63,64,65,66]. It is worth noting that with the trisomy 21 is associated the Down’s syndrome, which causes an approximately 20-fold increase in the risk of ALL. However, the rate of high hyperdiploidy has been shown to be significantly lower in Down syndrome patients with ALL compared to non-Down syndrome patients. Nevertheless, constitutional trisomy 21 still increases the risk of hyperdiploid ALL, which is characterized by the same chromosomal gains [55,67]. It is also worth noting that it has been suggested that congenital aneuploidy in B-ALL may be related to chromosomal instability, which may relate to telomere dysfunction through progressive telomere shortening or telomere aggregation [68]. Telomere dysfunction predisposes to malignant transformation in the human hematopoietic compartment, through mechanisms that may promote not only the appearance of structural chromosomal aberrations, but also the appearance of numerical aberrations [69]. Nevertheless, additional research will determine the relationship between these mechanisms.

When discussing this subtype, attention should also be paid to genetic variants in *CEBPE*, *ARID5B*, *PIP4K2A* and *BMI*, which are more strongly associated with high hyperdiploid ALL compared to the others [45,46,70,71]. Moreover, high hyperdiploidy is often found in patients with rare germline *ETV6* mutations and Noonan syndrome. This indicates the possibility of an inherited predisposition to develop a subtype of high hyperdiploidy in ALL. This was investigated by Smith et al., who assessed the frequency of predisposing germline mutations in 57 patients with this subtype of ALL. Three of their patients had confirmed germline mutations, and six had putative predisposing mutations that were rare in unselected individuals. Another three patients carried rare and predicted deleterious germline mutations in *GAB2*. In their results, they confirmed the significant contribution of rare, high penetrance germline mutations to the etiology of high hyperdiploid ALL. Additionally, *GAB2* identified by them is one of the activators of PI3K/AKT signaling, which, when activated, plays a key role in preventing apoptosis. Therefore, GAB2-associated mutations may represent a novel ALL predisposing gene. Cells of this subtype of leukemia are particularly susceptible to apoptosis, which makes them one of the easiest to treat during induction therapy. For this reason, it seems that the *GAB2* mutation present in cancerous cells could improve their ability to survive. However, to be able to finally confirm this dependence, additional studies are required, in particular those conducted in familial leukemias [72,73].

Certain chromosomal enhancements also affect the patient’s results. In this context, the simultaneous gain of +4, +10 and +17 chromosomes have been shown to improve patient outcomes. This turned out to be so significant that it is now used by the Children’s Oncology Group (COG) as a very low risk factor for disease recurrence [28,44,56]. Similarly, favorable patient outcomes can be seen with high +18 hyperdiploidy, while chromosome 5 gains are likely associated with a poorer prognosis [40,57,74,75,76]. In addition, this was confirmed by the results of Ramos-Muntada et al., in whose study the percentage of trisomy 18 in diagnostic samples was significantly higher in patients with complete remission than in patients with relapse [16].

Additionally worth highlighting is the recent analysis of the prognostic profile of UKALL hyperdiploidy, which used two trial datasets: a discovery cohort (UKALL 97/99) and a validation cohort (UKALL2003). Karyotypes with +17 and +18 or +17 or +18 in the absence of +5 and +20 included a good risk profile. Additionally, its prognostic effect was independent of minimal residual disease [77]. Patients with a high-risk hyperdiploid good risk profile showed a better response to treatment (relapse rate 5%) compared to other patients with high hyperdiploidy at 10 years (relapse rate 16%). It is noteworthy that almost half of the cases in the low-risk group had a modal chromosome number of 51–53 and could be excluded by groups using a DNA content threshold of 1.16. A high similarity between the results of patients in the high-risk hyperdiploid group with patients with an intermediate cytogenetic profile was noted [77].

However, genetic predisposition is not the only predictor of patient outcomes. Low WBC counts and no evidence of extramedullary disease combined with the aforementioned age of 2 to 10 years and the early pre-B phenotype, may also indicate a favorable prognosis for patients [33,78,79,80,81,82]. It has been proven that in combination with these clinical features, the survival rate of patients is even 90% [42,43,83].

Currently children with this chromosomal disorder show favorable results when treated with standard chemotherapy regimens. This is evidenced by the study by Rana et al., in which patients with hyperdiploidy (27 children) achieved post-induction remission rates of 100% [84]. Similarly, it was 99.7% in the study by Dastugue et al. [85]. It is believed that this is related to the potentially greater sensitivity of hyperdiploid lymphocytes to the cytotoxic effects of methotrexate, which is a component of many currently used therapies [28,75,86,87].

However, even with such good remission statistics, relapse is still possible. Although late relapses have been reported to be rare, they can still occur several years after discontinuation of treatment [88]. Additionally, in a study by Norén-Nyström et al. showed that the diagnosis of myelofibrosis is associated with an increased risk of recurrence after treatment of high hyperidiploid ALL and the most frequently described location is the bone marrow, which is responsible for approximately 75%. Extramedullary implication involving the Central Nervous System (CNS) occurs in fewer cases. Noting that the overall survival of children with the subtype of high hyperdiploid ALL is higher than the corresponding event-free survival rates, it can be concluded that the treatment of relapses is generally effective [55].

In the treatment and recurrence discussion, mutations targeting genes encoding histone modifiers (CREBBP, WHSC1, SUV420H1, SETD2 and EZH2) and the RTK-RAS pathway (FLT3, NRAS, KRAS and PTPN11) should also be mentioned. It is significant that in recent studies, the presence of mutations in the RAS pathway has been linked to a possible recurrence of the disease. For this reason, it seems beneficial to develop new targeted therapies, which would undoubtedly improve the results of treatment of patients [63,64,89,90,91].

Moreover, some studies suggest that parental exposure to organic solvents or substances such as amphetamines, marijuana, and cocaine correlate with the presence of the aforementioned RAS mutation in children with ALL [92]. Although no parental exposure is confirmed to be specific to childhood hyperdiploid ALL, it is possible that these substances may predispose the child to this subtype of leukemia [63,93]. However, the opposite seems to be the effect of smoking by the father. This is indicated by the lower incidence of hyperdiploid ALL in prenatally exposed children. It is assumed that this may be related to the mutagenic effect of smoking, which in turn may be toxic to hyperdiploid cells. However, further research is undoubtedly needed to confirm these theses [93]. In the environmental context, noteworthy is the work of Hjalgrim et al., who first presented and then confirmed the hypothesis that children weighing more than 4 kg at birth are at higher risk of developing acute leukemia, including ALL. Additionally, they showed that birth weight was also a risk factor for high hyperdiploid ALL subtype as well as t(12;21)-positive ALL [94,95].

Compared to low hyperdiploidy, its high variety seems to be well described in the literature. This is due to the fact that it is one of the most common genetic subtypes of ALL, which clearly translates into the amount of research conducted on this subject. However, its pathogenetic impact is still poorly understood. For instance, recent studies indicate low expression of CTCF and cohesin in hyperdiploid ALL. As they are the main regulators of chromatin architecture, they can significantly influence the dysregulation of gene expression throughout the genome. However, further research is needed to confirm the prevalence and impact of this phenomenon in aneuploid leukemias [96]. 

A similar explanation also requires the issues of inheritance of predisposition to this subtype of leukemia. It is generally believed that hyperdiploidy is a prenatal event constituting a so-called “first hit” with eventual RAS mutations acquired as a “second hit” after birth. This is supported by a study by Davidow et al., who presented a case of monozygous monochorionic twins who developed concordant hyperdiploid B-ALL (53 chromosomes). In leukemic cells, identical chromosomal gains were seen, however, with different RAS mutations. Despite this, both this study and evidence from other clinical trials are based on relatively small groups of patients. It would be beneficial to obtain similar results from larger studies [97].

### 2.3. Near-Triploidy

Near-triploidy is uncommon in children with ALL with a frequency of 0.3–2.1% [47,48]. The median age of children in the near-triploidy group is 4.5 years [48]. The studies did not notice the dominance of one of the sexes [48]. At time of diagnosis, patients had a median leukocyte count of 11.5 × 10^9^/L and no evidence of mediastinal tumor or CNS involvement [48]. The estimated 5-year overall survival for patients in the near-triploid group (94%) is similar to that for patients in the high-hyperdiploid group (91%) [50]. The near-triploid karyotype has nonrandom gains of chromosomes X, 4, 5, 6, 8, 10, 11, 12, 13, 14, 15, 16, 17, and 21 [98]. The remaining chromosomes are trisomic or tetrasomic, with the most common tetrasomies affecting chromosomes 8, 10, 11, 14, 18, 21 and X [50]. Moreover, Pui et al. reported occurrence of chromosomal abnormalities in the near-tetraploid BCP-ALL group, such as pentasomy or hexasomy of chromosome 21. In addition, paired chromosomes containing the same rearrangement were reported in six out of nine patients [48]. The mutational landscape of near-triploid BCP-ALL is characterized mainly by the TP53 mutation [99]. One hypothesis suggested that hyperdiploidy is mainly due to the simultaneous growth of chromosomes in one abnormal mitosis [100]. Ohtaki et al. proposed several mechanisms to explain the formation of near-triploid karyotypes, including nondisjunction, duplication of hypodiploid cells, loss of chromosomes from tetraploid cells, and multipolar mitosis of tetraploid cells [101].

### 2.4. Near-Tetraploidy

In 1968, Sandberg et al. reported the first patients with ALL exhibiting near-tetraploid, despite many years it is still a poorly understood anomaly due to its rare occurrence (1.4%) [48,102]. Several studies have reported a higher proportion of males in the near-tetraploidy group [48,51]. Near-tetraploidy is more common in older children, with a median age at diagnosis of 8.6 years [48,51]. WBC count below than 15 × 10^9^/L in cases of near-tetraploidy BCP-ALL [51]. Patients with BCP-ALL with near-tetraploidy and near-triploidy are B-cell precursor immunophenotype with positivity for CD19, CD22 [48]. A mediastinal mass was present in 20% of patients and initial CNS leukemia in 10% patients [48]. Patients with a near-tetraploidy profile are characterized by frequent tetrasomies of chromosomes 5, 8, 10, 11, 12, 18, 19, 20, 21, and sex chromosomes. More than four copies are most often found on chromosomes 3, 5, 7, 8, 17, 19 or 21 [48]. The conducted studies have shown that pediatric near-triploidy and near-tetraploidy is associated with the ETV6eRUNX1 fusion in leukemic B-lineage cells [51]. The EFS for patients with near-triploid ALL is not significantly different from the EFS for patients with hyperdiploid ALL and is 89% after 5 years [50]. The genetic and biological factors underlying near-triploid and near-tetraploid B-ALL are not fully understood and require further research [49,50].

## 3. Hypodiploidy

Hypodiploidy is defined as the loss of one or more chromosomes and is a rare cytogenetic abnormality, occurring in 7% of all cases of B-ALL. Most hypodiploid B-ALLs contain 45 chromosomes (80%) and are classified as near-diploid ALLs [103,104]. In most studies and treatment protocols, hypodiploid B-ALL is strictly defined as ≤44 chromosomes and can be further subdivided based on chromosome number as: near-haploid with 24–31 chromosomes, low-hypodiploid with 32–39 chromosomes and high hypodiploidy with 40–44 chromosomes [18].

### 3.1. Near-Haploidy

Near-haploid occurs in approximately 0.5% of all B-ALL cases and is restricted to childhood B-ALL [38]. The age of the patient at the time of diagnosis is relatively low and the median age varies between studies, where in the Saint Jude Total Therapy Study 15&16 (SJTTS) the median is 3.6 years [105] and in the Medical Research Council (MRC) study it is 7.4 years [104]. A retrospective analysis by the Ponte di Legno Childhood ALL Working Group (PDLWG) [39] found a male to female ratio was 1.46 in 101 cases of near-haploidy, but subsequent studies of haploidy show about the same incidence in males and females [106,107].

Near-haploid is associated with white blood cell (WBC) counts of <50 × 10^9^/L [39]. By flow cytometry analysis, hypodiploid cells are B-cell precursor immunophenotype positive for CD19, CD22, CD34, cCD79a and TdT. In addition, near-haploid ALL is characterized by CD10 positive [108,109]. Furthermore, the T-ALL immunophenotype is rarely observed in children with hypodiploid ALL. It is generally TdT-positive and expresses T-cell specific markets: CD1a, CD2, CD3, CD5 and CD7 [38,110]. This was confirmed in a larger series of studies reported by Nachman et al. and the UK Medical Research Council found that only 1% of all cases with chromosome < 40 hypodiploidy had the T-cell immunophenotype, while 11–15% of highly hypodiploid cases had the T-ALL immunophenotype [111]. The French–American–British (FAB) cell morphology subtype L2/L1 or L2 is reported more frequently in patients with hypodiploid B-ALL, as confirmed in a series reported by the Children Cancer Group (CCG) where 25% of cases are of the L2 subtype [103,112,113]. Clinical features at diagnosis in patients with near-haploid ALL include extramedullary leukemic foci manifested by splenomegaly in 44% and hepatomegaly in 43%, respectively. Less common are mediastinal tumors (7%) or enlarged lymph nodes (30%). In the PDLWG study, testicular disease was reported in only 4% (2 out of 46) and CNS involvement in only 1% (1 out of 115) [39,113,114].

In near-haploid ALL, retention of disomies 8, 10, 14, 18, 21, X and Y can be observed [104,109,115]. The most commonly lost chromosomes are 3, 7, 9, 15, 16 and 17 [99,104,109,116]. In a study by Harrison et al., the 64% patients had 26 chromosomes [104]. In hypodiploid < 40 ALL, especially in near-haploid B-ALL karyotypes, structural chromosomal aberrations (i.e., chromosomal translocations) are rare; however, it then increases with more chromosomes present in a low-hypodiploid karyotype [104]. This may suggest that massive chromosomal loss alone may be enough for leukemogenesis and that unconserved random chromosomes may contain specific genes that increase the oncogenic potential of leukemic cells [117,118].

Near-haploid B-ALL has been shown to exhibit distinctive and diverse gene expression profiles in addition to massive genetic losses [106]. The most common genetic changes found in this subtype are RAS signaling (*NRAS*—15% of patients; *FLT3*—9% of patients; *KRAS*—3% of patients; and *PTPN11*—1.5% of patients) and receptor tyrosine kinase (RTK), which occurs in >70% of cases [41,109,116]. Furthermore, Holmfeldt et al. also reported various mutation profiles that are characteristic of near-haploid ALL, including *NF1* (44%), histone modifiers (64%), *CREBBP* (32%), *CDKN2A*/B (22%), histone gene cluster 6p22 (19%) %), *IKZF3* (13%) and *PAG1* (10%) [109]. Point mutations in *EP300* and *EZH2* are the least common (<5%) [41,109,116]. Near-haploid B-ALL is not well characterized in association with any known hereditary predisposition syndrome, but mutations such as the *PTPN11*, *NRAS* substitution have been reported in patients diagnosed with Noonan syndrome. Importantly, single-nucleotide variants (SNVs) are germline SNVs and in some cases this close haploid may be a manifestation of this cancer predisposing syndrome [105,109]. Moreover, Kurtz et al. reported a case of a patient with Rubinstein-Taybi syndrome with a constitutional heterozygous missense variant in the *CREBBP* gene who developed a near-haploid B-ALL poorly responding to conventional chemotherapy [119]. Reports by Mullighan et al. support speculation that patients with *CREBBP* lesions may be more susceptible to therapy failure and disease relapse, as *CREBBP* mediates the expression of glucocorticoid-responsive genes and may play an active role in glucocorticoid response [105]. Another mutation that has been associated with poorer outcomes is *PAG1* deletions with MRD positivity [109].

### 3.2. Low Hypodiploidy

The frequency of patients with low hypodiploidy increases with age and occurs in 0.5% of children and approximately 4% of adult patients [104]. In the pediatric population, patients with low hypodiploidy are more common in older children than those with near-haploidy, with a median age at diagnosis of 12.9 years [39]. Although low-hypodiploid ALL was initially reported to be more common in men, Moorman et al. in their study, showed that 52% of patients with low hypodiploid were men and reported about the same prevalence in men and women [38,104].

Patients in this subgroup have relatively low white blood cell counts of <50 × 10^9^/L at diagnosis [118]. This is confirmed by studies by Pui et al., where 66% of patients with low-hypodiploid status had WBC: ≤20 × 10^9^/L, 22.2% of patients had >50 × 10^9^/L [39]. These results are consistent with previous reports that 85% of those in the low-hypodiploid group had WBC: ≤20 × 10^9^/L and 4% of patients had WBC: >50 × 10^9^/L [104]. In the cases of low hypodiploidy, a more immature B-cell precursor immunophenotype is found that is consistent with the pro-B ALL phenotype [108,109]. Extramedullary leukemic lesions involving the spleen (28%) and liver (35%) are common in patients with newly diagnosed low-hypodiploid ALL. 13% of patients have a mediastinal tumor and 14% of patients have enlarged lymph nodes. As with the near-haploid, only 1% of patients with low hypodiploidy ALL have CNS involvement, defined as the presence of blasts in a CSF sample with ≥5 leukocytes/μL and <10 erythrocytes/μL or the presence of cranial nerve palsy. Summing up the clinical picture of B-ALL patients with near-haploid and low-hypodiploid, it seems clear that most of the clinical and biological features do not indicate such a poor prognosis [39,113,114].

The retained disomies in low hypodiploid mainly include 1, 5, 6, 8, 10, 11, 14, 18, 19, 21, 22, X, and Y [104,109,116]. Patients with low-hypodiploid B-ALL may be at higher risk for additional structural chromosomal aberrations. This thesis is confirmed by Nachman et al., who in their research reported that eight out of nine patients with structural abnormalities have less than 36 chromosomes [111]. However, more research is needed to acknowledge whether pediatric patients with additional structural changes are also older than those without them, consistent with several reports of adult B-ALL with structural chromosomal aberrations [116,120].

More than 90% of low-hypodiploid patients have been identified with *TP53* mutations, which occur in virtually all low-hypodiploid B-ALL cases due to the very recurrent loss of chromosome 17 [41,121]. In addition, 50% of non-cancerous cells in children with low-hypodiploid B-ALL are found to have a *TP53* mutation, these cases may be a manifestation of Li-Fraumeni syndrome or other germline *TP53* cancer-predisposing mutations [109,121,122]. The presence of germline *TP53* mutations is associated with an increased risk of disease recurrence and is associated with the development of secondary malignancies [39,123]. This was confirmed by a study by Winter et al., where they reported that germline *TP53* mutations were associated with an increased incidence of secondary malignancies after hematopoietic stem cell transplantation (HSCT), highlighting the importance of germline testing in low-hypodiploid B-ALL to assess HSCT and possible benefits for patients with less genotoxic treatment strategies [124]. Therefore, children with B-ALL with low hypodiploid who are carriers of the *TP53* mutation and their relatives are referred for genetic consultation [122,125]. Other characteristic and recurrent genetic mutations or deletions in the low-hypodiploid cases are *CREBBP* (60%), *RB1* (41.2%), *IKZF2* (HELIOS; 52.9%), and *CDKN2A*/2B (20%) [109,116]. Notably, patients with near-haploid and low-hypodiploid B-ALL show no recurrent mutations in genes associated with activation of the RAS pathway, and transcriptomic analysis of pathway activation in low-hypodiploid B-ALL revealed constitutive signaling of the RAS and PI3K pathways [109]. A study by the Japan Association Childhood Leukemia Study Group (JACLS) showed that five of nine patients in both near-haploid and low-hypodiploid B-ALL have a high mutation rate in CIC, which is a member of the high-mobility group (HMG) transcription repressor superfamily and is mutated multiple times in oligodendrogliomas and round cell sarcomas. However, further research is needed to determine whether CIC germline status may be more prevalent in Asian populations than in non-Asian populations [126].

Young patients with hypodiploid B-ALL with <40 chromosomes have particularly poor clinical outcomes, with event-free survival (EFS) rates of 25–40% for near-haploid ALL and 30–50% for low-hypoid ALL [38,40,127]. This is supported by the MRC UK-ALL study in children, which showed a 3-year EFS of 29% for hypodiploid below 40 chromosomes compared to 65% for hypodiploid above 40 chromosomes [104]. As reviewed by Groeneveld-Krentz et al. Pediatric B-ALL patients with hypodiploidy < 40 chromosomes show specific characteristics compared to other B-ALL groups, for example: shorter time to first relapse, more frequent high-risk assignment in the relapse study, and poorer second remission rates [29]. National Cancer Institute (NCI) analysis indicates that only measurable residual disease (MRD) at the end of induction (EOI) is a significant predictor of EFS in near-haploid and low-hypodiploid B-ALL groups [113]. Several groups escalated treatment based on MRD-based stratification protocols are associated with a better outcome after adjusting for sex, age, and leukocyte count, with a 5-year EFS of 62% [39,128,129]. Hematopoietic stem cell transplantation (HSCT) is often performed in first complete remission (CR1) and is traditionally reserved for pediatric patients with a very poor prognosis (EFS 50%), including patients with hypodiploidy (<44 chromosomes) [123,130]. However, recent reports by McNeer et al. and Pui et al. show that pediatric patients with hypodiploid B-ALL do not appear to benefit from CR1 HSCT whether the MRD is 0.01% or greater at the end of induction [39,114,116]. Nevertheless, new treatments or strategies based on comprehensive genetic testing are needed to improve the outcomes of children with hypodiploid B-ALL [15].

Case analyses of B-ALL with high levels of hyperdiploidy indicate that chromosome gains were the primary oncogenic event; therefore, similar pathogenic mechanisms involving large aneuploidy may occur in hypodiploidy [42,64]. This is also indicated by the study by Safavi et al., where eight samples in a near-haploid state and four in a low-hypodiploid state show that massive chromosome loss is the main oncogenic event and other oncogenic lesions occur after hypodiploidy [115]. Hypodiploidy is observed in a wide spectrum of cancers, further indicating that it is a major contributor to cancer [131]. It seems very interesting that both in near-haploid B-ALL and in high hyperdiploidy there is retention of similar chromosomes, especially 14, 18, 21 and X, that may play an important role in leukemogenesis in these cases [122]. Therefore, further research is very important to clarify the role and cooperation of genes on these chromosomes and their involvement in various cellular pathways, which will contribute to a better understanding of the pathogenetic mechanisms resulting from hypodiploidy in B-ALL. Gruhn et al. observed that pre-leukemic clones with various chromosome rearrangements and high levels of hyperdiploidy were present in cord blood samples and neonatal heel prick tests from patients who later developed B-ALL [132]. Despite the lack of direct evidence for B-ALL subtypes with a near-haploid and low-hypodiploid, it can be speculated that B-ALL hypodiploidy in children appears in utero during fetal hematopoiesis and acts as a pre-leukemic initiating event [133].

### 3.3. High Hypodiploidy

High hypodiploidy is a very rare anomaly occurring in approximately 4% of diagnosed cases of hypodiploidy in both children and adults, but with a predominance of the younger group [38,104]. In the pediatric population with hyperdiploid B-ALL, the median age at diagnosis is 4 years in both Asian and European populations [64,104,128]. Several studies in Asian countries have found higher proportions of women in groups with high hypodiploidy, in contrast to the European population, where there is a higher proportion of men [41,104,128]. Patients with chromosomes 40–44 seem to have more in common with the 45-chromosome group than those with less than 40 chromosomes [103,106]. In high-hypodiploid ALL, high incidence of complex chromosomal abnormalities, with a higher prevalence in chromosomes 7, 9, and 12 can be observed [104]. Safavi et al. in their studies showed that genetic alterations involve CDKN2A and TP53, which are also found in low hypodiploid [41]. These results suggest that high hypodiploid may be genetically similar to low hypodiploid, but a limitation is the small number of high hypodiploidy B-ALL patients [41]. On the other hand, high hypodiploidy in patients with B-ALL is associated with a better prognosis compared to <40 chromosomes. In the analysis by Pui et al., induction treatment with MRD-based stratification protocols was associated with an outcome as follows: 5-year event-free survival rates of 75% [39].

### 3.4. Masked Hypodiploidy

It is quite common for B-ALL patients with near-haploid and low-hypodiploid status to have a clone with an exact or near-exact chromosome doubling of the hypodiploid clone, resulting in a clone with a modal chromosome number of 50–78 in the hyperdiploid range [38,111]. Various studies report chromosome doubling in 64% of nearly haploid cases and 44% of low-hypodiploid cases [134,135]. The double clone contains tetrasomies of those chromosomes that were disomic in the hypodiploid clone and a disomy of those that were monosomic, leading to complete loss of heterozygosity (LOH) in these chromosomes [99,111,117,134]. Moreover, it has been noticed that there is a preferential loss of chromosomes after doubling, and most often these are chromosomes: 2, 5, 6, 10, 14 and 22 [116]. It is worth emphasizing that the duplicated clone may be the only one detected at the time of diagnosis, i.e., the so-called masked hypoploidy [38,111,134]. This may result in misclassification to a lower risk group and treatment for hyperdiploid ALL B, which may result in treatment failure [15,104,108,111]. Studies have reported that there are not significant differences in clinical outcomes between patients with “masked hypodiploidy” and patients with mosaic double clone and hypodiploid clone, or those who have only hypodiploid clone [38,111,115,120,134]. So far, the molecular mechanism leading to chromosome doubling in nearly haploid and low-hypodiploid B-ALL has not been fully elucidated, since this phenomenon is quite common, research should be conducted to clarify the underlying mechanism [136,137,138].

## 4. Conclusions

In recent years, significant progress has been made in the field of ALL diagnostics, but many questions remain unanswered. As presented in our review, the same disease can have very different genetic variations. Undoubtedly, further publications describing the association of ALL with high hyperdiploidy and other related features could speed up its detection. Developing personalized treatment options based on targeted therapies and new patients’ stratification pathways could further improve patient outcomes. The common feature of the other aneuploidy subtype is their poor understanding due to the small number of patients and they should be given no less attention because these groups are associated with a worse prognosis than high hyperdiploidy. Therefore, it is important that future studies focus on elucidating the etiology and pathogenesis of the disease, which will result in new targeted therapies and improve the prognosis not only of ALL patients, but also of all hematological diseases.

## Figures and Tables

**Figure 1 ijms-24-08764-f001:**
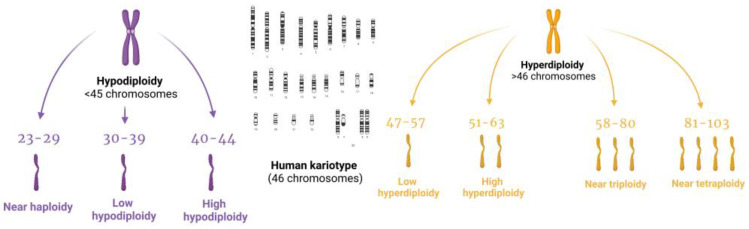
Schematic illustration of the new division of leukemia subtypes based on the WHO-HAEM5.

**Table 1 ijms-24-08764-t001:** Division of the ALL subtypes according to the ploidy of the chromosomes affected on the basis of the 2020 International System of Cytogenetic Nomenclature classification [17,22].

Subtype of the ALL	Number of Affected Chromosomes
Near-haploidy	23–29
Low hypodiploidy	30–39
High hypodiploidy	40–44
Correct number	46
Low hyperdiploidy	47–50
High hyperdiploidy	51–57
Near-triploidy	58–80
Near-tetraploidy	81–103

## Data Availability

No new data were created or analyzed in this study. Data sharing is not applicable to this article.

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
