# Peer review of "Overview on Aneuploidy in Childhood B-Cell Acute Lymphoblastic Leukemia"

_ijms, 2023, doi:10.3390/ijms24108764_

Round 1

Reviewer 1 Report

In this review, Panuciak et al described the intimate relationship between aneuploidy and treatment responses in childhood B-cell acute lymphoblastic leukemia. Regarding the impact of aneuploidy in the response to the treatment as well as to the overall survival, the aim of this review is highly important in term of mechanisms as well as of clinical settings.The paper is well organized and it can be considered as an update in the topic. 

However, the paper is very long and some paragraphs need to be rewritten in order to simplify the message. The implicated chromosomes in the aneuploidy (loss or gain) can be added in the tables 1 or 2. The part of ethnicity in the aneuploidy of this pathology can be presented in a separate paragraph with other confounding factors of this disease. 

The conclusion needs additional data concerning the application of these aneuploidy data in the management of not only ALL patients but also in all hematological diseases. The possible role of telomere dysfunction in this aneuploidy as well as in the relationship between aneuploidy and chromosomal instability can be discussed.   

The authors can give more importance for some language mistakes. 

Author Response

Dear Sir or Madam, thank you very much for the review of the manuscript. 

In response to your comments, we would like to thank you for appreciating our manuscript. Thus, you can find paragraphs which involve changes in the corrected manuscript.

In this review, Panuciak et al described the intimate relationship between aneuploidy and treatment responses in childhood B-cell acute lymphoblastic leukemia. Regarding the impact of aneuploidy in the response to the treatment as well as to the overall survival, the aim of this review is highly important in term of mechanisms as well as of clinical settings.The paper is well organized and it can be considered as an update in the topic.

However, the paper is very long and some paragraphs need to be rewritten in order to simplify the message.

-It has been corrected.

The implicated chromosomes in the aneuploidy (loss or gain) can be added in the tables 1 or 2.

-It has been added.

 The part of ethnicity in the aneuploidy of this pathology can be presented in a separate paragraph with other confounding factors of this disease.

-It has been moved.

The conclusion needs additional data concerning the application of these aneuploidy data in the management of not only ALL patients but also in all hematological diseases.

-It has been added.

 The possible role of telomere dysfunction in this aneuploidy as well as in the relationship between aneuploidy and chromosomal instability can be discussed.

--It has been added.

Reviewer 2 Report

Dear authors, I found your paper well written and complete. I only have a suggestion: please decribe better your methodology for selecting  (exclusion and inclusion criteria) the papers that you included in your analisys. 

Here other two issues that should be consedered:

The Authors decided to adopt the International System of Cytogenetic Nomenclature classification for literature review but a rationale for this decision is not clear given, especially in light of the already available WHO-HAEM5 and ICC classifications. Are there discrepancies between these classifications? If so, the Authors should address this point, give an explanation of their classification choice, and comparatively review the various cytogenetic classification systems.

Hyperploidy schema in Figure 1 reports overlapping figures (chromosomes) between low, high hyperploidy and near triploidity

Author Response

Dear Sir or Madam, thank you very much for the review of the manuscript.

In response to your comments, we would like to thank you for appreciating our manuscript. Thus, you can find paragraphs which involve changes in the corrected manuscript.

Dear authors, I found your paper well written and complete. I only have a suggestion:

please decribe better your methodology for selecting  (exclusion and inclusion criteria) the papers that you included in your analisys.

--It has been added and corrected.

Here other two issues that should be consedered: The Authors decided to adopt the International System of Cytogenetic Nomenclature classification for literature review but a rationale for this decision is not clear given, especially in light of the already available WHO-HAEM5 and ICC classifications. Are there discrepancies between these classifications? If so, the Authors should address this point, give an explanation of their classification choice, and comparatively review the various cytogenetic classification systems.

--It has been added.

Hyperploidy schema in Figure 1 reports overlapping figures (chromosomes) between low, high
hyperploidy and near triploidity.

-Correction applied.

Reviewer 3 Report

The review from Panuciak, Nowicka and colleagues is an overview of the aneuploidy often observed in acute lymphoblastic leukemia (ALL) and is based on the recent update of WHO-HAEM5.  The authors also relate the various chromosome aberrations to patient outcome.

The review is quite comprehensive and well written.

Minor Points:

The authors use the term B-ALL throughout the review, but do they actually mean BCP-ALL to describe the childhood leukemia subtypes that they describe?  (They do use this term on line 58 but not again). The term B-ALL is still used to describe mature B-cell leukemia, such as Burkitts’, and indeed adult leukemia.  To make the review more clear to readers this abbreviation should be checked throughout and changed where necessary e.g., hyperdiploidy should really say BCP-ALL.

What is the role of chromosome / microsatellite instability?

Figure 1. could be bigger / better.

I am also not sure that Table 1 really offers any more than what is written in the main text?

Author Response

Dear Sir or Madam, thank you very much for the review of the manuscript.

In response to your comments, we would like to thank you for appreciating our manuscript. Thus, you can find paragraphs which involve changes in the corrected manuscript.

The review from Panuciak, Nowicka and colleagues is an overview of the aneuploidy often observed in acute lymphoblastic leukemia (ALL) and is based on the recent update of WHO-HAEM5.  The authors also relate the various chromosome aberrations to patient outcome.

The review is quite comprehensive and well written.

Minor Points:

The authors use the term B-ALL throughout the review, but do they actually mean BCP-ALL to describe the childhood leukemia subtypes that they describe?  (They do use this term on line 58 but not again). The term B-ALL is still used to describe mature B-cell leukemia, such as Burkitts’, and indeed adult leukemia.  To make the review more clear to readers this abbreviation should be checked throughout and changed where necessary e.g., hyperdiploidy should really say BCP-ALL.

-Correction applied.

What is the role of chromosome / microsatellite instability?

--It has been added.

Figure 1. could be bigger / better.

-Correction applied. Please refer to the uploaded separate file with Figure 1. Its shape and design do not allow it to be enlarged specifically in the text due to the imposed text formatting restrictions.

I am also not sure that Table 1 really offers any more than what is written in the main text?

  • Thank you for your comment. The table illustrates what is written in the text, collecting information in one place and helping to visualize the described data. Given that the other two reviewers did not make a similar comment, after discussion, we decided to keep Table 1.

Round 2

Reviewer 1 Report

Some modifications have been added to the manuscript.  It can be accepted in the present form 

minor mistakes